# Molecular Insights into the Role of Sterols in Microtuber Development of Potato *Solanum tuberosum* L.

**DOI:** 10.3390/plants13172391

**Published:** 2024-08-27

**Authors:** Lisset Herrera-Isidron, Eliana Valencia-Lozano, Braulio Uribe-Lopez, John Paul Délano-Frier, Aarón Barraza, José Luis Cabrera-Ponce

**Affiliations:** 1Unidad Profesional Interdisciplinaria de Ingeniería Campus Guanajuato (UPIIG), Instituto Politécnico Nacional, Av. Mineral de Valenciana 200, Puerto Interior, Silao de la Victoria 36275, Guanajuato, Mexico; buribel1900@alumno.ipn.mx; 2Departamento de Ingeniería Genética, Centro de Investigación y de Estudios Avanzados del IPN, Unidad Irapuato, Irapuato 36824, Guanajuato, Mexico; eliana.valencia@cinvestav.mx (E.V.-L.); john.delano@cinvestav.mx (J.P.D.-F.); 3CONAHCYT-Centro de Investigaciones Biológicas del Noreste, SC., Instituto Politécnico Nacional 195, Playa Palo de Santa Rita Sur, La Paz 23096, Baja California Sur, Mexico; abarraza@cibnor.mx

**Keywords:** potato, *Solanum tuberosum*, microtubers (MTs), sterol biosynthesis, transcriptome sequencing, qRT-PCR, darkness, cytokinin

## Abstract

Potato tubers are reproductive and storage organs, enabling their survival. Unraveling the molecular mechanisms that regulate tuberization is crucial for understanding how potatorespond to environmental stress situations and for potato breeding. Previously, we did a transcriptomic analysis of potato microtuberization without light. This showed that important cellular processes like ribosomal proteins, cell cycle, carbon metabolism, oxidative stress, fatty acids, and phytosterols (PS) biosynthesis were closely connected in a protein–protein interaction (PPI) network. Research on PS function during potato tuberization has been scarce. PS plays a critical role in regulating membrane permeability and fluidity, and they are biosynthetic precursors of brassinosteroids (BRs) in plants, which are critical in regulating gene expression, cell division, differentiation, and reproductive biology. Within a PPI network, we found a module of 15 genes involved in the PS biosynthetic process. Darkness, as expected, activated the mevalonate (MVA) pathway. There was a tight interaction between three coding gene products for HMGR3, MVD2, and FPS1, and the gene products that synthetize PS, including CAS1, SMO1, BETAHSD, CPI1, CYP51, FACKEL, HYDRA1, SMT2, SMO2, STE1, and SSR1. Quantitative real-time polymerase chain reaction (qRT-PCR) confirmed the expression analysis of ten specific genes involved in the biosynthesis of PS. This manuscript discusses the potential role of genes involved in PS biosynthesis during microtuber development.

## 1. Introduction

Plant sterols, called phytosterols (PS), are integral components of the plant cell membranes. They are mainly found in the plasma membrane and in lower amounts in tonoplasts, mitochondria, and outer membranes of chloroplasts. PS forms lipid microdomains, providing fluidity, permeability, and transport across the membranes [1,2,3]. They are involved in signal transduction [4], auxin and ethylene signaling, the auxin efflux carrier system [5], regulation of cellulose biosynthesis [6,7], cell wall development [8], cyclin CycD3 activation [9], endocytosis, and cytoskeleton maintenance [10]. They are also essential in early embryo development [11,12,13], vascular patterning [14], stem elongation [15], plastid development [16], tillering [17], modulation of flowering, and potato tuberization [18,19].

Sitosterol is the most prevalent PS in *Arabidopsis thaliana* seedlings, followed by campesterol, stigmasterol, and more than 20 other minor PS, many of which are metabolic intermediates [12]. Sitosterol and stigmasterol are significant components of the plasma membrane (PM), which are critical for membrane fluidity and permeability [20,21]. Campesterol acts as a precursor for many brassinosteroids (BRs), which promote the growth of stems by increasing their length and encouraging cell division [20,21].

The initial genetic evidence supporting the involvement of PS in plant growth and development came from three *Arabidopsis* mutants that had impaired PS production: C-14 reductase (*fk*) [12,13], cephalopod/sterol methyl transferase 1 (*cph/smt1*) [1,22], and sterol C-8,7 isomerase (*hyd1*) [5]. It was found that all three enzymes act upstream of the BRs DWARF (DWF) pathway and have the inability to save the mutants by providing BRs, which led to the concept that PS are involved in new signaling pathways [23].

Overexpression of *StHMGR1*, *StHMGR3*, *StHMGR1*/*StFPS1*, and *StHMGR3*/*StFPS1* in potatoresulted in changes in the expression of PS biosynthesis genes, which are the primary end-products of cytosolic and plastidial isoprenoids (PS, SGAs, chlorophylls, and carotenoids). Additionally, this overexpression affected flowering, stem height, biomass, and tuber weight [18]. Double mutant *fps1/fps2* are embryo lethal [24]. The transcript levels of two distinct flowering locus (FT) paralogs, *SELF-PRUNING StSP3D* (associated with the transition to flowering) and *StSP6A* genes (associated with the induction of tuberization), were elevated in these lines, suggesting that both *StHMGR* and *StFPS* play a role in regulating the transitions of flowering and tuberization in potato [18].

Zhu et al. 2019 [25] have shown that exogenous epibrassinolide spraying in potato plants up-regulates the expression of the main regulators of tuberization, *StCDF1* and *StSP6A*, 8- and 4-fold, respectively, and increased the number and fresh weight of potato tubers. Silencing the *StBRI1* (*BRASSINOSTEROIDS INSENSITIVE 1*), the functional receptor in potato for BRs signaling, showed a decreased expression level of *StSP6A* and, consequently, a reduced number of tubers and yield [19].

Potato(*Solanum tuberosum* L.) contain bioactive lipids, including PS, in addition to carbohydrates, nutrients, and minerals [26,27,28]. PS are synthesized via the mevalonate (MVA) pathway, which occurs in the cytoplasm [29,30]. PS have been extensively studied due to their health-promoting effects. However, the primary focus in potato plants has been on steroidal glycoalkaloids (SGA), which are metabolites produced from PS and possess undesirable dietary qualities such as a bitter taste and toxicity [31].

We have previously reported a transcriptome analysis of potato microtuberization under darkness and osmotic stress conditions [32]. Further data analyses revealed, using a protein–protein interaction (PPI) network, that within the transcriptome analysis, a gene module of PS biosynthesis interacts with carbon metabolism, fatty acid biosynthesis, the cell cycle, and ribosomal proteins. To elucidate the molecular mechanisms involved during potato microtubers (MTs) development, we analyzed MVA-related PS biosynthesis and tuberigen regulatory genes, highlighting its regulation by cytokinin and darkness.

These findings provide novel approaches to enhance the quality of tubers by addressing the role of PSs’ response to environmental stress. This will be advantageous for breeding programs and genetic editing.

## 2. Results

### 2.1. Microtuber Development

Potato seedlings of the *Solanum tuberosum* cv. Alpha MTs variety were produced by culturing stolon explants in a growth medium consisting of MS media with 8% sucrose, 6 g/L of gelrite, 10 mg/L of 2-iP, and 0.3% w/v activated charcoal—the incubation period lasted for three weeks in darkness, according to Valencia-Lozano et al. (2022) [32] (Figure 1a,b). The MTs in Figure 1a were 4 mm in diameter, with a white to yellowish color, and developed in basal and upward in the stolon explants after three weeks in culture. Figure 1b shows mature MTs that are 7–9 mm in diameter and brown-colored after 2 months in culture. Figure 1c shows stolon control explants that were grown in MS medium that had 1% sucrose, 3 g/L of gelrite, 10 mg/L of 2-iP, and 0.3% w/v activated charcoal added to it. After three months in culture, we never observed microtuberization under these conditions.

### 2.2. Identification of Up-Regulated Transcripts Involved in PS Biosynthesis

To obtain an overview of the potato gene expression profile during the tuberization period, cDNA samples from tubers were subjected to RNAseq and transcriptomic analyses. Sequencing of cDNA libraries generated a total of 397,834,274 reads. Of these, 4756 transcript genes were significantly (>2-fold-change and *p* < 0.05) and differentially expressed. A total of 2896 transcript genes were up-regulated, and 1860 transcript genes were down-regulated (Figure 2a) [32]. It is remarkable that from the 4756 differentially expressed genes, 21 transcript genes (19 up-regulated and 2 down-regulated) correspond to MVA-related and PS biosynthesis pathway genes (Figure 2b). Moreover, we proceeded to compare the MVA-related and PS biosynthesis pathway gene expression levels between MTs induced in dark conditions and grown plants in control conditions from the normalized reads to TPM (transcripts per million). The hierarchical bi-clustering analysis was used to confirm the DEG analysis and to obtain a better perspective of the analyzed data (Figure 3).

To determine the interaction among the MVA-related and PS pathway gene products, a PPI network analysis was carried out with stringent parameters to ensure the robustness and confidence of the resulting protein interactions network. The confidence value was set at 0.800, which ensured a high degree of likelihood since, from the 19 DEGs of the MVA-related and PS biosynthesis genes, only 15 were kept in the resulting PPI network (Table 1). The modules of the PPI network found in this analysis are as follows: Ribosome biogenesis, cell cycle, carbon metabolism, starch biosynthesis, glycerophospholipid metabolic process, fatty acid metabolism, histone modification, reactive oxygen species (ROS), phenylpropanoid biosynthesis, PEBP family members, and PS and MVA biosynthesis pathways (Figure 4). Remarkably, the gene products of the MVA-related or PS biosynthesis pathway exhibited the highest degree of interaction, suggesting a tight regulation explicitly based on each biosynthetic pathway (Appendix A, Figure 5).

Finally, we selected the genes *FSP1* and *HMGR3* from the MVA-related pathway, *CPI1*, *CYP51*, *FACKEL*, *SI1/HYD1*, *SMT1*, and *SMT2* from the PS biosynthesis pathway, and *StSP6A* and *FD* from the tuberigen regulatory pathway to determine whether the transcriptional behavior’s observed trend in the transcriptomic analysis is consistent with the expression levels assessed through quantitative PCR (qPCR) analysis. The qPCR analysis of the selected genes involved in the MVA-related PS biosynthesis and tuberigen regulation pathways showed an up-regulation trend, except for *SMT1*, which exhibited a down-regulation behavior, and altogether have been shown the same behavioral expression trend for both analytical approaches: transcriptomics and qPCR (Figure 6).

## 3. Discussion

PS are essential components of cellular membranes, signal transduction, regulation of plant ontogenesis, and stress tolerance. To understand the roles of PS in the potato microtuberization process under darkness, we performed data mining of the complete transcriptomics data set to focus on the MVA-related and PS biosynthesis pathway-related gene transcripts activity. Overall, 21 DEGs for MVA-related and PS biosynthetic pathway transcript genes were identified in the transcriptomics data set of MTs induced under dark conditions (Figure 2). A PPI analysis network of MVA-related and PS biosynthetic pathway genes with a high stringency threshold (0.800) value was performed, in which 15 of the 21 DEGs gene products were kept, 3 gene products (HMG1/HMGR3, MVD2, and FPS1) belonged to the MVA pathway and 12 gene products (CAS1, SMO1, 3BHSD/D2/HSD1, CPI1, CYP51, FACKEL, SI1/HYD1, SMT2, SMO2, STE1/DWF7, SSR1,bAS) highlighted a high degree of regulation in the PS biosynthesis in the microtuberization process under the darkness condition (Figure 4 and Figure 5, Table 1). Furthermore, the cytosolic MVA pathway is involved in the isoprenoids, PS (campesterol, stigmasterol, and sitosterol), and plant hormones biosynthesis (abscisic acid, BRs, cytokinins, gibberellic acid, and strigolactones) [33]. The PPI network analysis showed that Hydroxy-3-methylglutaryl CoA reductase (*HMG1*, M1CAE9_SOLTU 3, *PGSC0003DMT400063296*) interacts with MVD2 (M1C7S0_SOLTU, PGSC0003DMT400061600) and FPS1 (M1CX22_SOLTU, PGSC0003DMT400076602) (Figure 5). Moreover, this gene was up-regulated ~4-fold (2.013 Log2-scale) with respect to the control conditions (Figure 5 and Figure 2b). HMG1/HMGR3 catalyzes the conversion of HMG-CoA to MVA [34] and three *HGMR* genes have been characterized in the cultivated potato species [35]. It is worth noting that MVA biosynthesis is activated in darkness and is negatively regulated by light, as well as the transcriptional activities of HMG1/HMGR3, CAS1, and SMT1. However, HMG1/HMGR3 transcriptional activity is also induced in darkness, as we have shown in our microtuberization protocol and the respective transcriptomics data set obtained [32,36,37,38]. Moreover, *HMG1* is directly involved in flowering regulation, fruit size regulation in early development, cellular development and proliferation, and defense response activation against pathogens [18,39,40,41]. Dwarf potato plants were produced by the overexpression of *StHMGR3*, with unbalanced PS levels (cycloartenol, sitosterol, and stigmasterol), and were rescued by the overexpression of *StFPS1/StHMGR3*. Potato plants were phenotypically normal, with delayed flowering, higher total biomass, and tuber weight [18].

MVD2 executes the initial step in the production of isoprene-containing compounds, including PS and terpenoids. It is also a rate-limiting enzyme in the MVA pathway that synthesizes isopentenyl diphosphate (IPP) from mevalonate 5-diphosphate, which is the universal precursor of isoprenoids (IPP) [42,43]. In turn, FPS1 interacts with CAS1 (M1CST0_SOLTU, PGSC0003DMT400073861) (Figure 4), which carries out the conversion of 2,3-oxidosqualene to cycloartenol, leading this compound to the main PS synthesis (campesterol, sitosterol, and stigmasterol) in higher plants [44]. Furthermore, CAS1 is directly involved in the regulation of chlorophyll synthesis, cell division cycle regulation, cellular proliferation, and the regulation of cellular growth, development, and differentiation [16].

CAS1 interacts with SMO1 (M1AYT4_SOLTU, PGSC0003DMT400033236) and HSD1/3BHSD/D2 (M1AB90_SOLTU, PGSC0003DMT400018853) (Figure 5). SMO1 carries out the production of 4a-carboxy-sterol and is directly involved in the embryogenesis process regulation, cellular differentiation, cell cycle division, and establishment of shoot apical meristem [45,46]. HSD1/3BHSD/D2 carries out the synthesis of 3-oxosteroid and is responsible for the allosteric regulation of this biosynthetic process [45].

SMO1 also interacts with CPI1 (M1A0D5_SOLTU, PGSC0003DMT400011907) (Figure 5). CPI1 is a key enzyme that catalyzes the breaking of the 9β,19-cyclopropane ring of the 4α-methyl-cyclopropylsterol cycloeucalenol, resulting in the formation of the Δ8-sterol obtusifoliol [47]. CPI is directly involved in the regulation of the cellular differentiation process, gametophyte formation, stomal differentiation and development [48], chlorophyll allocation, and root gravitropism [49].

In turn, CPI1 interacts with CYP51 (M1BCS3_SOLTU, PGSC0003DMT400042257) (Figure 4 and Figure 5). CYP51 is a member of the cytochrome P450 monooxygenases super-family and regulates an essential step in plant PS biosynthesis and has remained very well conserved throughout the evolution of eukaryotes [50,51]. CYP51 is directly involved in cell membrane formation and grain and fruit development [52], flowering process regulation, and pollen fertility [53]. Moreover, CYP51 interacts with FACKEL/FK (M0ZSD2_SOLTU, PGSC0003DMT400007043) and SI1/HYD1 (SI1, PGSC0003DMT400071184) (Figure 4 and Figure 5). FACKEL is directly involved in the regulation of embryo development [12,13], the cellular differentiation process, and regulates cytokinin and ethylene signaling [54]. CYP51 interacts with SI1/HYD1 (Figure 4 and Figure 5). SI1/HYD1 conducts the synthesis of 24-methylenelophenol, is directly involved in root and vascular growth [14], differentiation, patterning, and development [55], and drives the polar auxin transport machinery [5].

Furthermore, SI1/HYD1 interacts with SMT2 (M1BLQ0_SOLTU, PGSC0003DMT400047969) and SMO2 (M0ZQ08_SOLTU, PGSC0003DMT400005519) (Figure 4 and Figure 5).

SMT2 is an important regulatory branch point to yield 24-ethylidenlophenol and sitosterol [56] and regulates the concentration ratios among 24-methyl and 24-ethyl sterol [57,58], playing an important role in the regulation of cell growth [12], embryogenic development [13], and proliferation. Additionally, it is involved in responding to abiotic stress and has a direct impact on BRs synthesis [59].

SMO2 carries out the synthesis of Δ7-avenasterol and plays a crucial role in both the embryonic and postembryonic development processes and auxin homeostasis [60].

Also, SMO2 interacts with STE1/DWF7 (M1CI11_SOLTU, PGSC0003DMT400067881) (Figure 4 and Figure 5). STE1 mediates the synthesis of 5-dehydroepisterol and 5-dehydroavenasterol [61,62] and is directly involved in the regulation of cell proliferation and elongation and BR synthesis regulation [62].

Finally, STE1/DWF7 interacts with SSR1/DWF1 (SSR1/DWF1, PGSC0003DMT400054476) (Figure 4 and Figure 5). SSR1/DWF1 performs the synthesis of the precursor of the phytohormone brassinolide [63] and, in turn, regulates the brassinolide content, accumulation, and allocation, and regulates cellular proliferation, elongation, and differentiation [64,65].

From the data set analyzed, the DEG and qPCR transcriptional expression levels of the essential regulatory genes of the MVA-related and PS biosynthesis pathway, coupled with the PPI, all together converge in the regulatory model establishment of MTs development under a high content of sucrose/gelrite, 2-iP, and darkness conditions, as shown in Figure 7. We have found that a mix of sterols (sitosterol, stigmasterol, and campesterol) at 200 μM increases both the number and size of MTs. Furthermore, after three weeks of incubation in darkness, we observed an inhibition of MTs when we used an inhibitor of sterol biosynthesis, terbinafine, at 5 mg/L. Taken all together, sterols play a crucial role during the microtuberization process. In this model, we have analyzed the genes *FACKEL*, *HYDRA1*, *SMT1*, *CYP51*, and *SMO1*, which are expressed in the shoot apex meristem to differentiate a tuber. *FACKEL*, *HYDRA1*, *SMT1*, *SMT2*, *SSR1*, *STE1*, and *SMO1* activate a further differentiation of the tuber internal tissues, leading to the development of cambium, xylem, phloem, and starch storage parenchyma. *HYDRA1 and CPI1* promote the development of protoderm and ground meristems.

## 4. Materials and Methods

### 4.1. Plant Material and MTs Induction

MTs induction was carried out according to Valencia-Lozano et al. (2022) [32]. In brief, potato *S. tuberosum* cv alpha stolon explants were cultured in the induction medium MR8-G6-2-iP (8% w/v sucrose, 6 g/L of gelrite, 10 mg/L of 2-iP, and 0.3% w/v activated charcoal). The control medium utilized was MR1-G3-2-iP (1% w/v sucrose, 3 g/L of gelrite, 10 mg/L of 2-iP, and 0.3% w/v activated charcoal). Containers were kept in darkness at 25 °C/17 °C for 15 days. The MTs were then collected, frozen in liquid nitrogen, and subsequently stored at −80 °C for preservation. The study employed five biological replicates for both the treatment and control groups.

### 4.2. Isolation of RNA, qPCR, and Transcriptome Sequencing

Total RNA was isolated using the Trizol reagent (Invitrogen, Carlsbad, CA, USA). The RNA concentration was measured by its absorbance at 260 nm, and the 260 nm/280 nm absorbance ratio was evaluated. The RNA integrity was verified through 2% (w/v) agarose gel electrophoresis. The cDNA samples were amplified with SYBR™ Green (ThermoFisher CAT: 4312704, Waltham, MA, USA) in Real-Time PCR Systems (CFX96 BioRad, Hercules, CA, USA).

The cDNA samples were sequenced using the Illumina HiSeq 4000 platform (Illumina, San Diego, CA, USA). The quality of the sequenced reads was checked with the FastQC v0.11.3 software package (http://www.bioinformatics.babraham.ac.uk/projects/fastqc/; accessed on 15 March 2023), and the reads were processed to remove sequence adapters and low-quality bases using Trimmomatics (v0.39; accessed on 16 March 2023) [66].

### 4.3. Analysis of DEG and Interaction Analysis of PS Enzymes

The RNAseq reads were aligned to the *S. tuberosum* reference genome available in Phytozome v12.1. (https://phytozome.jgi.doe.gov/pz/portal.html; accessed on 15 March 2023) using the STAR aligner v.2.5.2b [67]. This process generated BAM (Binary Alignment/Map) files, and transcript read counts were obtained using the featureCounts program from the Subread v.1.5.2 package [68]. The DESeq2 v1.12.4 tool was used to estimate the differentially expressed genes [69], and the transcript ontology analysis study was conducted via Blast2GO. V6.0 (https://www.blast2go.com/; accessed on 8 April 2023).

A gene network was constructed using STRING [70] with a confidence score of 0.800, incorporating homologous genes from the *S. tuberosum* genome in the Sol Genomics Network. Gene identifiers (Id) were based on the UNIPROT [71] and NCBI databases [72]. Protein sequences in *S. tuberosum* that have a similarity of above 60% with *A. thaliana* were considered.

### 4.4. Transcriptional Analysis through qPCR of Genes Involved in PS Biosynthesis

The set of genes for qPCR analysis is outlined in Table 2. *EF1* and *SEC3* expression levels were used as references to determine the relative expression of target genes using the 2^−∆∆CT^ method [73]. Five biological replicates were analyzed for each sample, with three technical replicates performed for each biological replicate during qPCR analysis.

## 5. Conclusions

The MTs’ development was enhanced by osmotic stress, 2-iP supplementation, and darkness conditions, and the mechanism might be attributed to the orchestration of the MVA-related and PS biosynthesis genes pathways as follows: *HMG1/HMGR3* and *FPS1* (MVA-related pathway) activate the expression of the tuberigen *StSP6A* and the transcription factor FD, thereby causing potato tuber formation.

The MVA-related pathway is activated under darkness, providing MVA and IPP to balance the major plant PS (sitosterol, stigmasterol, and campesterol). PS biosynthetic gene modulations of *FACKEL, SI1/HYD1, SMT1, CYP51,* and *SMO1* orchestrate and promote the establishment of the architecture of the apical meristem in the first developmental stages of tuber differentiation and in the vascular tissue’s development. *SI1*/*HYDRA1* and *CPI1* positively regulate the protoderm and the development, positively influencing the microtuberization enhancement process of potato.

## Figures and Tables

**Figure 1 plants-13-02391-f001:**
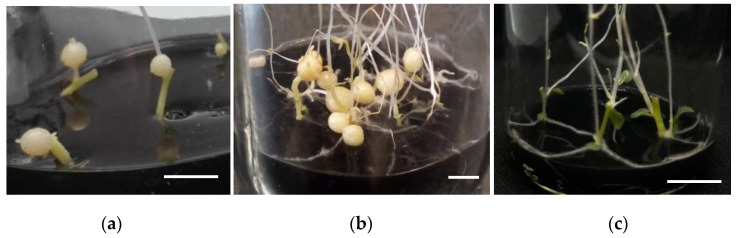
Potato *Solanum tuberosum* L. cv Alpha MTs development. (**a**) MTs development from stolon explants cultured in MR8-G6-2-iP medium after three weeks of incubation in darkness. The scale bar represents 1.0 cm. (**b**) Mature MTs development from stolon explants in MR8-G6-2-iP medium after two months of incubation in darkness. Scale bar represents 1.0 cm. (**c**) Stolon explants cultured in control conditions: MS medium, 1% sucrose, 3 g/L gelrite, 10 mg/L 2-iP, and 0.3% w/v activated charcoal in darkness. The scale bar represents 0.5 cm.

**Figure 2 plants-13-02391-f002:**
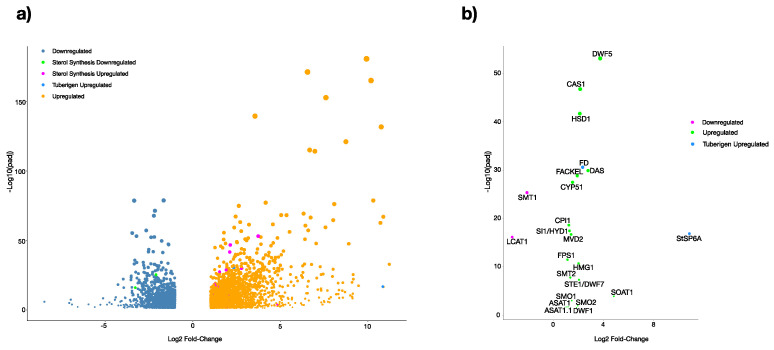
Volcano plot of differentially expressed genes (DEGs) in potato MTs induced in dark conditions. (**a**) DEG complete data set (>2-fold change and *p* < 0.05). (**b**) DEG of MVA-related and PS biosynthesis pathway genes.

**Figure 3 plants-13-02391-f003:**
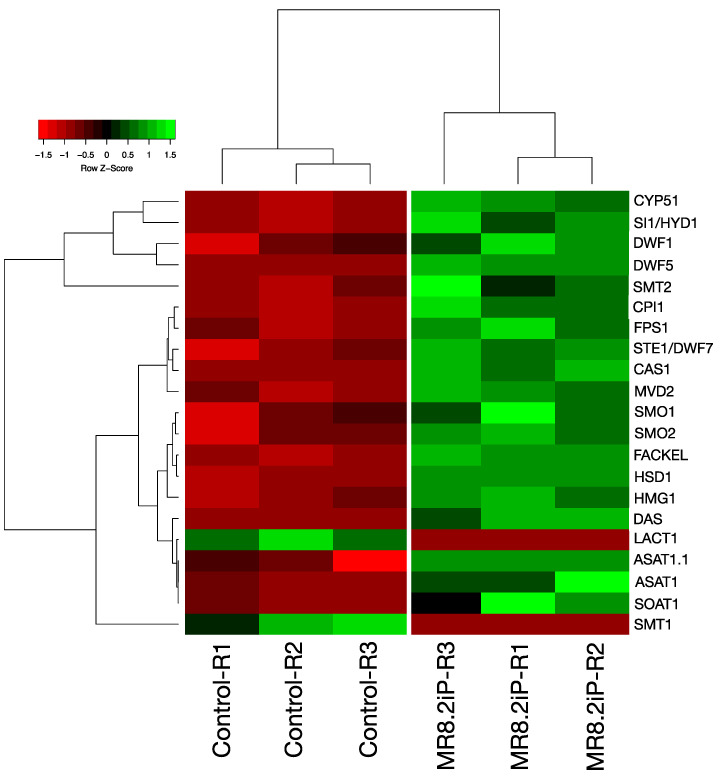
Hierarchical clustering analysis (HCA) and heat map displaying the up-regulated genes involved in PS biosynthesis during potato MTs development in darkness; the up-regulation levels are shown in Log2.

**Figure 4 plants-13-02391-f004:**
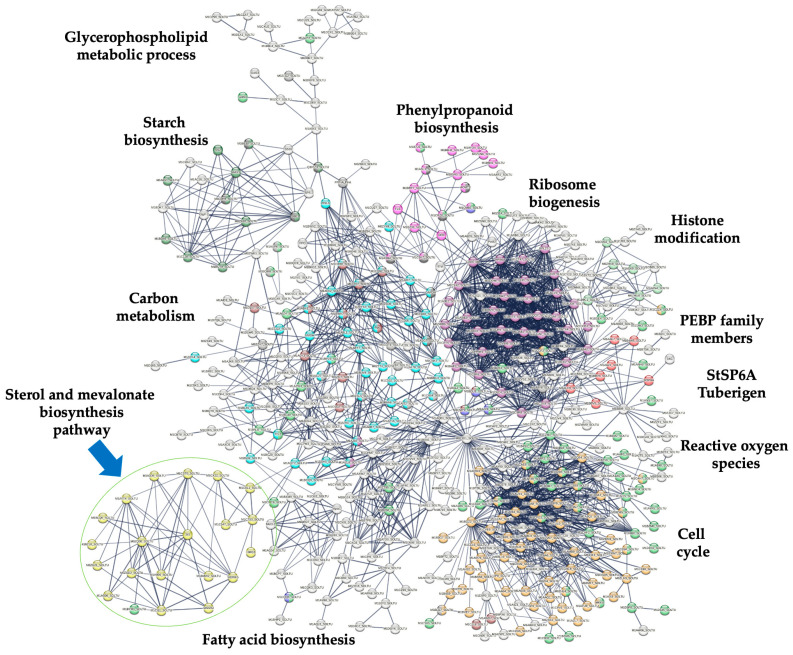
PPI network of up-regulated genes derived from the STRING database v12.0 for potato *S. tuberosum*, based on transcriptomic-wide analysis with high confidence (0.800). Modules are highlighted and blue arrow indicates the PS biosynthetic module. The figure illustrates a full network, where edges represent both functional and physical protein associations.

**Figure 5 plants-13-02391-f005:**
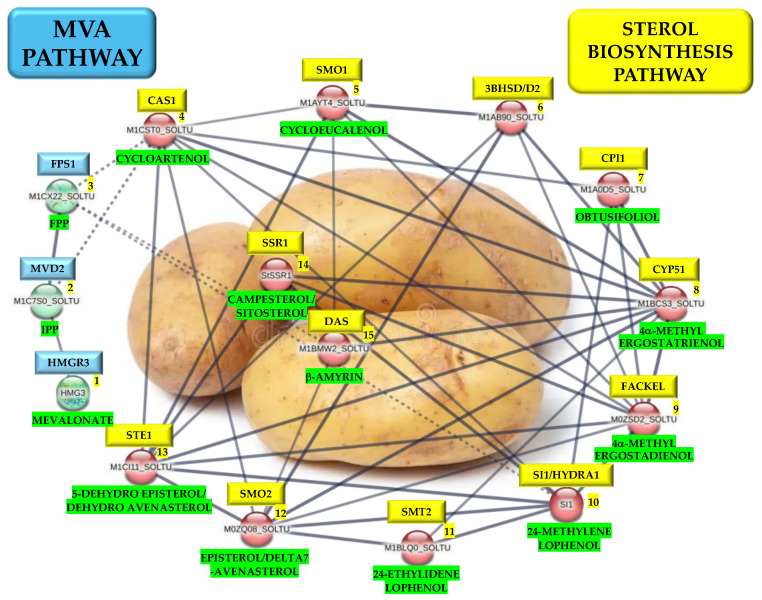
PPI network of up-regulated PS biosynthetic genes during potato tuberization, yielded in STRING database v12.0 with 0.800 confidence. Blue color represents genes involved in the MVA pathway, yellow represents PS biosynthetic genes, and green represents the products of each gene.

**Figure 6 plants-13-02391-f006:**
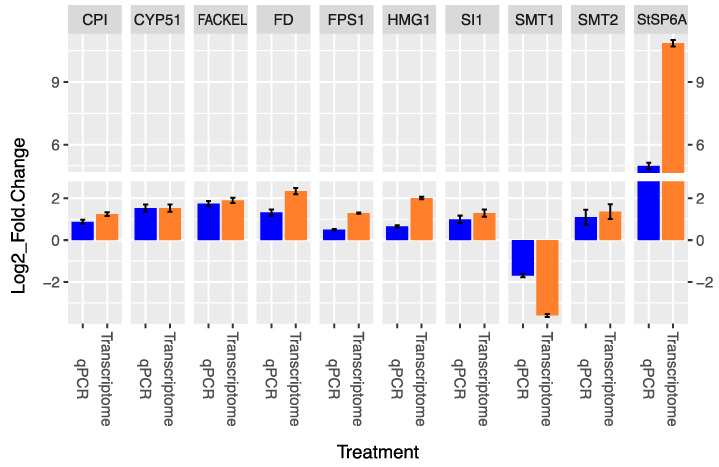
Quantitative RT-qPCR analysis validation of the MVA-related (two genes) and PS biosynthesis (six genes) and tuberigen regulatory (two genes) pathway showed significant differential expression levels through transcriptomics analysis. Blue bars correspond to relative expression estimation through qPCR analysis. Orange bars correspond to the DEG estimated relative expression through transcriptomics analysis. Relative expression estimation levels are represented in Log2-fold change.

**Figure 7 plants-13-02391-f007:**
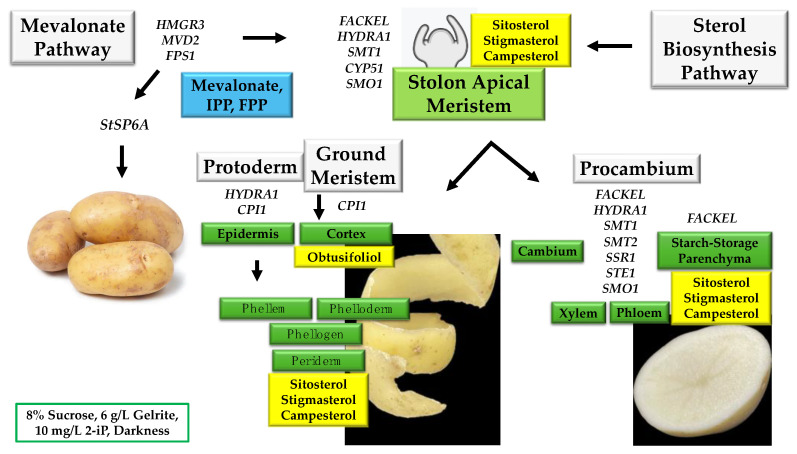
Potato *Solanum tuberosum* L. MTs development is enhanced by the modulation of the MVA and PS biosynthesis pathways in osmotic stress, 2-iP supplementation, and darkness conditions.

**Table 1 plants-13-02391-t001:** PS biosynthesis up-regulated genes identified in this work.

No	ID String v.12	*Arabidopsis*	Annotation
1	HMG3	*HMGR3*	Hydroxy-3-methylglutaryl-coenzyme A reductase 1-like
2	M1C7S0_SOLTU	*MVD2*	Diphosphomevalonate decarboxylase-like
3	M1CX22_SOLTU	*FPS1*	Farnesyl pyrophosphate synthase 1-like
4	M1CST0_SOLTU	*CAS1*	Cycloartenol synthase
5	M1AYT4_SOLTU	*SMO1-1*	Fatty acid hydroxylase domain-containing protein
6	M1AB90_SOLTU	*HSD1*	3βHSD domain-containing protein
7	M1A0D5_SOLTU	*CPI1*	Cycloeucalenol cycloisomerase
8	M1BCS3_SOLTU	*CYP51*	Sterol 14-demethylase
9	M0ZSD2_SOLTU	*FACKEL*	Delta14-sterol reductase
10	SI1	*HYD1*	Cholestenol Delta-isomerase
11	M1BLQ0_SOLTU	*SMT2*	24-methylenesterol C-methyltransferase
12	M0ZQ08_SOLTU	*SMO2*	Fatty acid hydroxylase domain-containing protein
13	M1CI11_SOLTU	*STE1/DWF7*	Delta7-sterol 5-desaturase
14	StSSR1	*DWF1*	Delta (24)-sterol reductase-like
15	M1BMW2_SOLTU	*DAS*	Delta-amyrin synthase

**Table 2 plants-13-02391-t002:** Primer-set from PS biosynthetic process DEGs used to validate their expression levels.

ID String v.12	ID	NCBI	Forward	Reverse
*M1BCS3_SOLTU*	*CYP51*	XM_006348474.2	CATACAGGCAAGGCAGAGAA	AGAGCAGCAATCAGAAGACC
*M1C7G1_SOLTU*	*FD*	XM_006361882.2	GAAAGCAGGCTTACACGAATG	GAAGTAGAGCACCAGCTGAA
*M0ZSD2_SOLTU*	*FACKEL*	XM_006341108.2	AAGACGTGTGGGCTGAATAC	TACGGTACGGTGAGGCTAAT
*M1CX22_SOLTU*	*FPS1*	XM_006344841.2	GGAGGTGTACTCTGTGCTTAAA	GATAGTCCTCGATTCAGCTTCC
*SI1*	*SI1*	NM_001288433.1	CACATGGTCCTTGAGGGATATT	CTCGGCCTACGTATCTTGAATC
*M1A996_SOLTU*	*SMT1*	XM_015310447.1	CTAAACCGTATTGGCGGAATTG	GGGCATGACATGTAGCTTCTA
*M1BLQ0_SOLTU*	*SMT2*	XM_006364446.2	CCCAATGGATTCTCTCACTCTC	GCCCGTTTACCCTTAACCT
*StSP6A*	*StSP6A*	XM_006365395.2	GGGAAACCTTTGGCAATGAAG	CCCAATTGCTGGAACAACAC
*M1A0D5_SOLTU*	*CPI1*	XM_015309779.1	GCAACACCTAGTCTGTGGTTAG	CACAACACCAAGGCACAAAG
*HMG3*	*HMG1*	XP_015159610.1	GCGACCTGTTAAGCCTCTATAC	CAACCCAGTGGTTAGGTACAA

## Data Availability

Data are contained within the article and Appendix A.

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
