# Peer review of "Molecular Insights into the Role of Sterols in Microtuber Development of Potato Solanum tuberosum L."

_plants, 2024, doi:10.3390/plants13172391_

Round 1

Reviewer 1 Report

Comments and Suggestions for Authors

The manuscript describes Molecular insights into the role of sterols in microtuber development of potato. The methods are common, but it could provide some useful information. It could be accepted after major and careful revision.

1.     Figure 1, add the control picture, and pictures in two different stages at least. The whole Text need add more detailed explaination.

2.     In 5.1, which variety of potato need be written out?

3.     In 2.1, the text is explained too simple? It need show two conditions (stress and control)? And related pictures?

4.     In 5.4 and Figure 6, they need perform significant difference analysis through t-test.

5.     Please add sterols content changes in different condition and stages.

6.     The format of the references?

Author Response

Answers to Reviewer 1 Comments

The manuscript describes Molecular insights into the role of sterols in microtuber development of potato. The methods are common, but it could provide some useful information. It could be accepted after major and careful revision.

  1. Figure 1, add the control picture, and pictures in two different stages at least. The whole Text need add more detailed explanation.

Dear Reviewer 1, Thank you very much for your interesting suggestions. We have added a new picture showing control and two stages of microtuberization. Explanations were also changed within the manuscript:

2.1. Microtuber Development

Potato seedlings of the Solanum tuberosum cv. Alpha MTs variety were produced by culturing stolon explants in a growth medium consisting of MS media with 8% sucrose, 6 g/L gelrite, and 10 mg/L of 2-iP. The incubation period lasted for three weeks in darkness, according to Valencia-Lozano et al. (2022) [32] (Figures 1A and B). MTs in Figure 1A were 4 mm in diameter, white to yellowish-coloured developed in basal and upward in the sto-lon explants, after three weeks in culture. Figure 1B shows mature MTs that are 7-9 mm in diameter and brown-colored after 2 months in culture. Figure 1C shows stolon control ex-plants that were grown in MS medium that had 1% sucrose, 3 g/L gelrite, and 10 mg/L 2-iP added to it. After three months in culture, we never observed microtuberization under these conditions.

Figure 1. Potato Solanum tuberosum L. cv Alpha MTs development. (A) MTs development from stolon explants cultured in MR8-G6-2-iP medium after three weeks of incubation in darkness. Scale bar represents 1.0 cm. (B) Mature MTs development from stolon explants in MR8-G6-2-iP medium after two months of incubation in darkness. Scale bar represents 1.0 cm. (C) Stolon explants cultured in control conditions: MS medium, 1% sucrose, 3 g/L gelrite, 10 mg/L 2-iP and 0.3% activated charcoal in darkness. Scale bar represents 1.0 cm.

  1. In 5.1, which variety of potato need be written out?

Dear Reviewer 1, Thank you for pointing it out. Information on the potato variety was included in section 5.1.

  1. In 2.1, the text is explained too simple? It need show two conditions (stress and control)? And related pictures?

Dear Reviewer 1, we agree with this comment. The correction was made as explained in comment No. 1 above (Section 2.1).

  1. In 5.4 and Figure 6, they need perform significant difference analysis through t-test.

Dear Reviewer 1, Thank you very much for your suggestion. We made a new Figure 6, including the standard error (section 5.4).

Figure 6. Quantitative RT-qPCR analysis validation of the MVA-related (two genes) and PS biosyn-thesis (six genes) and tuberigen regulatory (two genes) pathway showed significant differential expression levels through transcriptomics analysis. Blue bars correspond to relative expression es-timation through qPCR analysis. Orange bars correspond to the DEG estimated relative expres-sion through transcriptomics analysis. Relative expression estimation levels are represented in Log2-fold change.

  1. Please add sterols content changes in different condition and stages.

Dear Reviewer 1, Thank you very much for your comment, we quantified the sterol content in potato microtubers in mature microtubers according to Lieberman-Burchard methods, being of 52.7903±0.34 μg/Fw.

We have also made experiments adding sterol mixtures within the induction medium (sitosterol, stigmasterol and campesterol) 200 uM, and found increased number (0.5 vs 1 MT/explant) and diameter (3 mm vs 5 mm) of developed microtubers after three weeks of incubation. These results are shown in the following figure:

  1. The format of the references?

Dear Reviewer 1, the format of the references was made according to the Plant style.

Diener, A.C.; Li, H.; Zhou, W.; Whoriskey, W.J.; Nes, W.D.; Fink, G.R. STEROL METHYLTRANSFERASE 1 Controls the Level of Cholesterol in Plants. Plant Cell 2000, 12, 853–870, doi:10.1105/tpc.12.6.853.

Reviewer 2 Report

Comments and Suggestions for Authors

The authors propose a manuscript titled “Molecular insights into the role of sterols in microtuber development of potato Solanum tuberosum L.”. In this manuscript, authors performed a bioinformatic analysis on sterol biosynthesis in the morphogenesis of potato microtuber development, and found a module of 15 genes involved in the sterols biosynthesis. The results are important to understanding molecular mechanisms of potato tuber development.

In general, the manuscript is well organized and English is good. However, some revisions are required before publication in Plants. Most important points for the revision of the manuscript are:

1.Abstract: the abstract need to rewrite, and a concise abstract, including background, key methods, main results and scientific significance, is suggested.

2. It is suggested that the author conduct an expression pattern analysis for all or part of the 15 screened genes to further verify their express sites in MT. In addition, it is better to measure the content of sterols in relevant components of MT at different developmental stages.

3. Some sentences, such as in Lines 39-47, 173, 182, 266, need to rewrite.

Author Response

Answers to Reviewer 2 Comments

The authors propose a manuscript titled “Molecular insights into the role of sterols in microtuber development of potato Solanum tuberosum L.”. In this manuscript, authors performed a bioinformatic analysis on sterol biosynthesis in the morphogenesis of potato microtuber development and found a module of 15 genes involved in the sterols biosynthesis. The results are important to understanding molecular mechanisms of potato tuber development.

In general, the manuscript is well organized and English is good. However, some revisions are required before publication in Plants. Most important points for the revision of the manuscript are:

  1. Abstract: the abstract need to rewrite, and a concise abstract, including background, key methods, main results and scientific significance, is suggested.

Dear reviewer 2, Thank you very much for your suggestion, we modified the abstract as shown below:

Abstract: The potato tuber are reproductive and storage organs, enabling its survival. Unraveling the molecular mechanisms that regulate tuberization is crucial for understanding how potatoes respond to environmental stress situations and for potato breeding. Previously, we did a transcriptomic analysis of potato microtuberization without light. This showed that important cellular processes like ribosomal proteins, cell cycle, carbon metabolism, oxidative stress, fatty acids and phytosterols (PS) biosynthesis were closely connected in a protein-protein interaction (PPI) network. Research on PS function during potato tuberization has been scarce. PS play a critical role in regulating membrane permeability and fluidity, and they are biosynthetic precursors of brassinosteroids (BRs) in plants, which are critical in regulating gene expression, cell division, differentiation, and reproductive biology. Within a PPI network, we found a module of 15 genes involved in the PS biosynthetic process. Darkness, as expected, activated the mevalonate (MVA) pathway. There was tightly interaction between three coding genes products for HMGR3, MVD2, and FPS1, and the genes products that synthetize PS, including CAS1, SMO1, BETAHSD, CPI1, CYP51, FACKEL, HYDRA1, SMT2, SMO2, STE1, and SSR1. Quantitative real-time polymerase chain reaction (qRT-PCR) confirmed the expression analysis of ten specific genes involved in the biosynthesis of PS. This manuscript discusses the potential role of genes involved in PS biosynthesis during microtuber development.

  1. It is suggested that the author conduct an expression pattern analysis for all or part of the 15 screened genes to further verify their express sites in MT.

Dear reviewer 2, we appreciate a lot your appointment. We did not make this specific analysis yet; it is important to integrate the information in this way.

 In addition, it is better to measure the content of sterols in relevant components of MT at different developmental stages.

Dear Reviewer 2, Thank you very much for your comment. We quantified the sterol content in potato microtubers in stage 2 (mature microtubers) according to the Lieberman-Burchard method, being 52.79030.34 52.7903±0.34 ug/Fw.

We have also made experiments adding sterol mixtures within the induction medium (sitosterol, stigmasterol, and campesterol) 200 mM, and found an increased number (0.5 vs. 1 MT/explant) and diameter (3 mm vs. 5 mm) of developed microtubers after three weeks of incubation. These results are shown in the following figure:

  1. Some sentences, such as in Lines 39-47, 173, 182, 266, need to rewrite.

Dear Reviewer 2, we agree with this comment. We have modified the following sentences:

Line 39-47:

Plant sterols, called phytosterols (PS), are integral components of the plant cell membranes. They are mainly found in the plasma membrane and in lower amounts in tonoplasts, mitochondria, and outer membranes of chloroplasts. PS form lipid microdomains, providing fluidity, permeability, and transport across the membranes [1-3]. They are involved in signal transduction [4], auxin and ethylene signaling, auxin efflux carrier system [5], regulation of cellulose biosynthesis [6-7], cell wall development [8], cyclin CycD3 activation [9], endocytosis, and cytoskeleton maintains [10]. They are also essential in early embryo development [11-13], vascular patterning [14], stem elongation [15], plastid development [16], tillering [17], modulation of flowering, and potato tuberization [18, 19].

Line 173:

Furthermore, the cytosolic MVA pathway is involved in the isoprenoid’s, sterol (campesterol, stigmasterol, and sitosterol), and plant hormones biosynthesis (abscisic acid, brassinosteroids, cytokinins, gibberellic acid, and strigolactones) [33].

Line 182:

HMG1/HMGR3 catalyzes the conversion of HMG-CoA to mevalonate [34], and three HGMR genes have been characterized in the cultivated potato species [35]. It is worth noting that MVA biosynthesis is activated in darkness and negatively regulated by light, as well as the transcriptional activity of HMG1/HMGR3, CAS1, and SMT1. How-ever, the HMG1/HMGR3 transcriptional activity is also induced in darkness, such as we have been shown in our microtuberization protocol and the respective transcriptomics data set obtained [32,36-38].

Line 266:

The MVA-related pathway is activated under darkness, providing Mevalonate and IPP, to balance of the major plant sterols (sitosterol, stigmasterol and campesterol).

Reviewer 3 Report

Comments and Suggestions for Authors

This study was focused on the role of sterols in the tuberization of the potato plant by don conducting transcriptome sequencing. The authors found 15 genes that are involved in the biosynthesis of the sterols and, confirmed the expression of ten specific genes, and analyzed the protein-protein interactions of the related genes. Detailed methods were provided. The conclusions were supported by the results. I would recommend its acceptance for publication if the authors would address several issues listed below:

1. For the introduction part, first, the structure of the introduction part may be improved: in line 76, the "phytosterols" was mentioned again as it has appeared in line 38, the beginning of the introduction, please re-arrange the text; second, please provide sufficient information, for example, in line 82, please explain "MT".

2. Results presentation may be improved. First, the figures may not be boxed by dark edges; second, please adjust the font in figure2, 3, 5 etc to allow the readers to see the text well; Second, the figure 4 seems not clear enough, it is suggested that different group can be colored differently.

Comments on the Quality of English Language

The English language needs to be improved:

1. please avoid using too long sentences, for example, in line 39-45, 108-111, and line 221-225;

2. please check any grammar issue throughout the body text, for example, line 105-107, 188-189, 196-197, 266, etc

Author Response

Answers to Reviewer 3 Comments

This study was focused on the role of sterols in the tuberization of the potato plant by don conducting transcriptome sequencing. The authors found 15 genes that are involved in the biosynthesis of the sterols and, confirmed the expression of ten specific genes, and analyzed the protein-protein interactions of the related genes. Detailed methods were provided. The conclusions were supported by the results. I would recommend its acceptance for publication if the authors would address several issues listed below:

  1. For the introduction part, first, the structure of the introduction part may be improved: in line 76, the "phytosterols" was mentioned again as it has appeared in line 38, the beginning of the introduction, please re-arrange the text; second, please provide sufficient information, for example, in line 82, please explain "MT".

Dear Reviewer 3, Thank you very much for your suggestions, we have made corrections in the abstract for phytosterols (PS), as well as we homogenize the term sterol to PS throughout the document.

Abstract: The potato tuber are reproductive and storage organs, enabling its survival. Unraveling the molecular mechanisms that regulate tuberization is crucial for understanding how potatoes respond to environmental stress situations and for potato breeding. Previously, we did a transcriptomic analysis of potato microtuberization without light. This showed that important cellular processes like ribosomal proteins, cell cycle, carbon metabolism, oxidative stress, fatty acids and phytosterols (PS) biosynthesis were closely connected in a protein-protein interaction (PPI) network. Research on PS function during potato tuberization has been scarce. PS play a critical role in regulating membrane permeability and fluidity, and they are biosynthetic precursors of brassinosteroids (BRs) in plants, which are critical in regulating gene expression, cell division, differentiation, and reproductive biology. Within a PPI network, we found a module of 15 genes involved in the PS biosynthetic process. Darkness, as expected, activated the mevalonate (MVA) pathway. There was tightly interaction between three coding genes products for HMGR3, MVD2, and FPS1, and the genes products that synthetize PS, including CAS1, SMO1, BETAHSD, CPI1, CYP51, FACKEL, HYDRA1, SMT2, SMO2, STE1, and SSR1. Quantitative real-time polymerase chain reaction (qRT-PCR) confirmed the expression analysis of ten specific genes involved in the biosynthesis of PS. This manuscript discusses the potential role of genes involved in PS biosynthesis during microtuber development.

Line 82:

We have previously reported a transcriptome analysis of potato microtuberization (MTs) development under darkness and osmotic stress conditions [32]. Further data analysis revealed using a protein-protein interaction (PPI) network, that within the transcriptome analysis, a gene module of sterol biosynthesis interacts with carbon metabolism, fatty acid biosynthesis, cell cycle and ribosomal proteins. To elucidate the molecular mechanisms involved during potato MTs development, we analyzed MVA-related, sterol biosynthesis and tuberigen regulatory genes, highlighting its regulation by cytokinin and darkness.

  1. Results presentation may be improved. First, the figures may not be boxed by dark edges; second, please adjust the font in figure2, 3, 5 etc to allow the readers to see the text well; Second, the figure 4 seems not clear enough, it is suggested that different group can be colored differently.

Dear Reviewer 3, Thank you for pointing this out. The following modifications were made:

  • The dark edges that framed the figures were removed.
  • The font of the figures 2, 3, and 4 has been increased.
  • The colors used in Figure 4 has been improved.

The English language needs to be improved:

  1. please avoid using too long sentences, for example, in line 39-45, 108-111, and line 221-225.

Dear Reviewer 3, we agree with this comment. We have modified the following sentences:

Line 39-47:

Plant sterols, called phytosterols (PS), are integral components of the plant cell membranes. They are mainly found in the plasma membrane and in lower amounts in tonoplasts, mitochondria, and outer membranes of chloroplasts. PS form lipid microdomains, providing fluidity, permeability, and transport across the membranes [1-3]. They are involved in signal transduction [4], auxin and ethylene signaling, auxin efflux carrier system [5], regulation of cellulose biosynthesis [6-7], cell wall development [8], cyclin CycD3 activation [9], endocytosis, and cytoskeleton maintains [10]. They are also essential in early embryo development [11-13], vascular patterning [14], stem elongation [15], plastid development [16], tillering [17], modulation of flowering, and potato tuberization [18, 19].

Line 108:

Moreover, we proceeded to compare the MVA-related and PS biosynthesis pathway gene expression levels between MTs induced in dark conditions and grown plants in control conditions from the normalized reads to TPM (transcripts per million). The hierarchical biclustering analysis was used to confirm the DEG analysis and to obtain a better perspective of the analyzed data (Figure 3).

Line 221-225:

In turn, CPI1 interacts with CYP51 (M1BCS3_SOLTU, PGSC0003DMT400042257) (Figures 4 and 5). CYP51 is a member of the cytochrome P450 monooxygenases super-family, regulates an essential step in plant sterol biosynthesis and has remained very well conserved throughout the evolution of eukaryotes [50,51]. CYP51 is directly in-volved in cell membrane formation, grain, and fruit development [52], flowering process regulation, pollen fertility [53].

  1. please check any grammar issue throughout the body text, for example, line 105-107, 188-189, 196-197, 266, etc

Dear Reviewer 3, Thank you for your suggestions. In order to improve the grammar, we modified the following sentences:

Line 105-107:

It is remarkable that from the 4756 differentially expressed genes, 21 transcript genes (19 up-regulated, and 2 down-regulated) correspond to MVA-related and sterol biosynthesis pathway genes (Figure 2b).

Line 188-189:

It is worth noting that mevalonate biosynthesis is activated in darkness and negatively regulated by light, as well as the transcriptional activity of HMG1/HMGR3, CAS1, and SMT1. However, the HMG1/HMGR3 transcriptional activity is also induced in darkness, such as we have been shown in our microtuberization protocol and the respective transcriptomics data set obtained [32,36-38].

Line 196-197:

MVD2 executes the initial step in the production of isoprene-containing compounds, including sterols and terpenoids. It is also a limiting enzyme in the MVA pathway that synthesizes isopentenyl diphosphate (IPP) from mevalonate 5-diphosphate, which is the universal precursor of isoprenoids (IPP) [42].

Line 266:

The MVA-related pathway is activated under darkness, providing Mevalonate and IPP, to balance of the major plant sterols (sitosterol, stigmasterol and campesterol).

Round 2

Reviewer 1 Report

Comments and Suggestions for Authors

No other suggestion.

Round 2

"No other suggestion".

We express our sincere gratitude for the time and effort you have dedicated to reviewing our manuscript. Your insightful feedback and detailed comments have been incredibly valuable in improving the quality of our work.

Reviewer 2 Report

Comments and Suggestions for Authors

The revised manuscript has been greatly improved in the Sections of methods and methods, as well as results. The issues that I was mostly concerned about have also been largely addressed. Some minor language errors still need to be corrected, but they should be relatively easy.

Round 2

"The revised manuscript has been greatly improved in the Sections of methods and methods, as well as results. The issues that I was mostly concerned about have also been largely addressed. Some minor language errors still need to be corrected, but they should be relatively easy.".

We sincerely appreciate the time and effort you have devoted to reviewing our paper. We appreciate your valuable feedback and comments, as they have been beneficial in enhancing our work.

Reviewer 3 Report

Comments and Suggestions for Authors

Thank you for addressing my comments in the original version of the manuscript and making the improvements. I still have several concerns: 

1. figure 1, the figures may be prepared better, for example, the background may be made totally dark.

2. figure 6, all the "treatment" is the same, thus the figure is not concise, please correct.

3. figure 7, the caption itself is not sufficient for the readers to understand this figure, please expand an add explanation for the details.

Comments on the Quality of English Language

1. Please revise any grammar errors. For example, in lines 211 and 217, "carry" should be "carrys".

2. line 315-316, in "...expressions levels were used as reference genes...", "genes" should be deleted.

3. line 208, "limiting enzyme" should be revised to "rate-limiting enzyme"

Round 2

Answers to Reviewer-3 Comments:

Thank you for addressing my comments in the original version of the manuscript and making the improvements. I still have several concerns: 

  1. figure 1, the figures may be prepared better, for example, the background may be made totally dark.

Dear Reviewer 3, Thank you very much for your observation about Figure 1. We have modified it.

  1. figure 6, all the "treatment" is the same, thus the figure is not concise, please correct.

Dear reviewer 3, Thank you for pointing it out. In this figure, all genes show a significant difference in level of regulation. We are not sure what you mean when you say that all genes are the same, given that each gene exhibits differences in both transcriptome and qPCR analysis.

  1. figure 7, the caption itself is not sufficient for the readers to understand this figure, please expand an add explanation for the details.

Dear Reviewer 3, Thank you very much for your insightful suggestion. We have added new information and further explanations for the model shown in Figure 7.

Comments on the Quality of English Language

  1. Please revise any grammar errors. For example, in lines 211 and 217, "carry" should be "carrys".

Agree. We have made the substitution of carry for ”carries”

  1. line 315-316, in "...expressions levels were used as reference genes...", "genes" should be deleted.

Agree. We have made the respective modification.

  1. line 208, "limiting enzyme" should be revised to "rate-limiting enzyme"

Agree. It was corrected.
